# Pathogenesis, Murine Models, and Clinical Implications of Metabolically Healthy Obesity

**DOI:** 10.3390/ijms23179614

**Published:** 2022-08-25

**Authors:** Yun Kyung Cho, Yoo La Lee, Chang Hee Jung

**Affiliations:** 1Department of Internal Medicine, Asan Medical Center, University of Ulsan College of Medicine, Seoul 05505, Korea; 2Asan Diabetes Center, Asan Medical Center, Seoul 05505, Korea; 3Asan Institute of Life Science, University of Ulsan College of Medicine, Seoul 05505, Korea

**Keywords:** cohort study, metabolic syndrome, murine models, obesity, pathophysiology

## Abstract

Although obesity is commonly associated with numerous cardiometabolic pathologies, some people with obesity are resistant to detrimental effects of excess body fat, which constitutes a condition called “metabolically healthy obesity” (MHO). Metabolic features of MHO that distinguish it from metabolically unhealthy obesity (MUO) include differences in the fat distribution, adipokine types, and levels of chronic inflammation. Murine models are available that mimic the phenotype of human MHO, with increased adiposity but preserved insulin sensitivity. Clinically, there is no established definition of MHO yet. Despite the lack of a uniform definition, most studies describe MHO as a particular case of obesity with no or only one metabolic syndrome components and lower levels of insulin resistance or inflammatory markers. Another clinical viewpoint is the dynamic and changing nature of MHO, which substantially impacts the clinical outcome. In this review, we explore the pathophysiology and some murine models of MHO. The definition, variability, and clinical implications of the MHO phenotype are also discussed. Understanding the characteristics that differentiate people with MHO from those with MUO can lead to new insights into the mechanisms behind obesity-related metabolic derangements and diseases.

## 1. Introduction

Obesity is related to a decreased life expectancy, mostly because obese individuals have a higher risk of developing type 2 diabetes, cardiovascular disease (CVD), and cancer [1,2]. It has become a major public health concern as its prevalence has been increasing globally. In addition to the implications of obesity at the individual level, the obesity pandemic may impose a substantial health burden on society.

However, not all obese people have an increased risk of mortality, suggesting that there is a subpopulation of healthy obese individuals, i.e., those with so-called “metabolically healthy obesity” (MHO) [3,4]. MHO is distinguished by the absence of metabolic disturbances, including insulin resistance, type 2 diabetes, hypertension, and dyslipidemia, commonly associated with a chronic inflammatory state [4,5]. In this paper, we compare the pathophysiology of MHO with that of metabolically unhealthy obesity (MUO), and then describe some animal models of MHO based on the physiology and features of this phenotype. We will also address the concept of MHO and its clinical consequences given the unstable nature of this phenotype.

## 2. Main Pathophysiology of MHO Versus MUO

The clinical and pathophysiologic aspects that have been associated with MHO and MUO are depicted in Figure 1. Despite the fact that the precise mechanism behind the development of MHO remains unknown, numerous plausible pathways have been proposed in both human and animal research. These include altered adipokine secretion, suppressed inflammation and fibrosis, and a particular distribution pattern of fat tissue types, such as the accumulation of more subcutaneous but less visceral and ectopic fat.

### 2.1. Body Composition and Fat Distribution

In addition to functioning as an energy reservoir, adipose tissue is a hormonally active organ with specific biochemical properties that influence metabolic pathways. White adipose tissue (WAT) and brown adipose tissue (BAT) are the two primary forms of adipose tissue (BAT). WAT is primarily responsible for energy storage, whereas BAT is primarily responsible for energy expenditure via non-shivering thermogenesis in the mitochondria [6]. WAT depots can be divided into subcutaneous adipose tissue (SAT), which is the adipose tissue beneath the skin, and visceral adipose tissue (VAT) lining internal organs [7]. The accumulation of abdominal, visceral, and ectopic fat leads to insulin resistance and an unfavorable cardiometabolic profile [8,9,10,11,12]. Individuals with MHO are distinguished by higher levels of SAT and lower levels of VAT, and decreased fat deposition in liver and skeletal muscle, compared with MUO subjects with comparable levels of total fat mass and BMI [13]. In contrast, widespread VAT deposition, as measured by computed tomography, is linked to MUO, which is characterized by hyperinsulinemia, glucose intolerance, and atherogenic dyslipidemia [13].

According to several studies, it is indisputable that increased visceral adiposity independently correlates with a higher cardiometabolic risk [14,15,16,17,18,19]. In contrast, the metabolic profile of subcutaneous fat is more favorable [14,20]. Using whole-body MRI and spectroscopy, the German Tübingen diabetes family study evaluated total, visceral, subcutaneous abdominal, gluteofemoral and leg fat mass, and liver fat content to distinguish between individuals with healthy and unhealthy obesity [20]. In this study, a lower proportion of gluteofemoral fat mass and leg fat mass was found to be the most reliable predictor of a metabolically unhealthy condition. Increased gluteofemoral and leg fat mass and higher insulin sensitivity and preserved insulin secretion determined a metabolically healthy status while increased visceral fat mass, increased subcutaneous abdominal fat mass, and a high liver fat content contributed to a metabolically unhealthy phenotype [20,21]. Another German study with 314 white people found that the proportion of liver fat is a significant predictor of metabolically benign obesity [20]. Non-alcoholic liver disease (NAFLD) is strongly associated with the metabolically unhealthy obesity phenotype, although some subtypes of NAFLD with a stronger hepatic genetic component are not associated with insulin resistance and cardiometabolic risk [22,23]. Collectively, assessment of body fat distribution may be the key to understanding the pathophysiology linking obesity, metabolic health, and cardiometabolic risk.

### 2.2. Alterations in the Adipokine Phenotype

Adiponectin is the most extensively researched adipokine associated with MHO. In both men and women, adiponectin, the most abundant protein released by adipose tissue, is inversely correlated with percentage body fat and directly correlated with insulin sensitivity [24]. Plasma adiponectin levels in patients with MHO are reported to be higher than those in patients with MUO [25,26,27,28]. The reasons behind the lower adiponectin levels in patients with MUO are unknown but may be related to the chronic hyperinsulinemia observed in MUO, which suppresses adiponectin production in adipose tissue [29,30], resulting in a positive feedback cycle where decreased adiponectin secretion causes insulin resistance and increased insulin resistance, in turn, causes decreased adiponectin levels [3]. In addition to adiponectin, Sanidasa et al. investigated the cardioprotective (omentin-1) and non-cardioprotective (visfatin, resistin, chemerin) hormones in MHO and MUO [31]. The cardioprotective adipokines omentin-1 [32,33] and adiponectin were found to be higher in cases of MHO, but the non-cardioprotective adipokines visfatin and resistin [32,34,35] were found to be lower. Mateusz Lejawa et al. [36] recently observed some differences in the adipokine profiles between MHO and MUO in the cohort from the Metabolic and Genetic Profiling of Young Adults with and without a Family History of Premature Coronary Heart Disease (MAGNETIC) study. According to their findings, adipsin is linked only to MHO and not to MUO. Furthermore, markers such as ghrelin and PAI-1 are solely associated with MUO not with MHO [36]. Further research is required to determine the precise involvement of those adipokines in the cardiometabolic fates of MHO and MUO.

### 2.3. Adipose Tissue Inflammation and Fibrosis

Chronic inflammation, particularly in adipose tissue, has been recognized as the main pathophysiology of obesity-related comorbidities, where insulin resistance has a crucial role [3,37,38]. Previous research has shown that people with MUO exhibit higher levels of circulating inflammation markers such as C-reactive protein, plasminogen activator inhibitor-1 (PAI-1), IL-6, and TNF-α, compared with those with MHO [39,40,41,42,43,44,45]. However, a few studies [46,47,48] provided contradictory results, with no difference between the two groups. Such discrepancies can be attributed to inconsistencies in the definitions of MHO and MUO, differences in the sets of markers studied, or the relatively small number of participants. It has been reported that patients with MUO, compared with those with MHO, have higher M1-like (proinflammatory) macrophages and proinflammatory CD4+ T cells in adipose tissue [39,49,50,51,52,53,54]. In addition to inflammation, fibrosis in adipose tissue has also been postulated as an obesity-linked pathology [3]. Obese patients exhibit higher expression levels of collagen I, IV, V, and VI genes, and increased fibrosis, notably pericellular fibrosis in adipose tissue [55,56,57,58,59]. Adipose tissue collagen gene expression and collagen levels are also inversely correlated with insulin sensitivity in obese patients and decrease with weight loss [60,61,62,63]. Recently, Jun Yoshino et al. showed that adipose tissue expression of connective tissue growth factor is positively correlated with body fat mass and inversely correlated with insulin sensitivity indices [64]. These findings corroborate the hypothesis that adipose tissue fibrosis is linked to MUO, as observed in animal models [65].

## 3. Animal Models of MHO

Obesity models using mice have provided invaluable insights into obesity in humans and associated metabolic consequences, such as metabolically protected obesity. In this section, we introduce genetic mouse models harboring some characteristics of MHO (Table 1).

### 3.1. Adiponectin Transgenic Mouse

Adiponectin is an anti-inflammatory, insulin-sensitizing adipokine expressed by adipocytes that improves lipid and glucose metabolism via several mechanisms. Kim et al. overexpressed adiponectin in transgenic ob/ob mice, resulting in a 2–3 fold increase in steady-state levels of adiponectin complexes in plasma [66]. These animals exhibit higher levels of PPAR-gamma target gene expression and lower levels of macrophage infiltration in adipose tissue and suppressed systemic inflammation [66]. As a result, the transgenic mice were morbidly obese, with considerably more adipose tissue than their ob/ob littermates, resulting in an intriguing paradox of increased fat mass paired with improved insulin sensitivity [66]. Overexpression of adiponectin results in the development of hyper-obese animals that exhibit subcutaneous fat as the most abundant type of adipose tissue, a larger number of adipocytes with a much smaller average cell size, reduced inflammation, and metabolic fitness, all of which are specific features of MHO [66].

### 3.2. Txnip Knockout Mice

Thioredoxin-interacting protein (Txnip), a cellular oxidative stress regulator, is activated by hyperglycemia and limits glucose absorption into fat and muscle tissues. Txnip knockout mice acquire considerably greater adipose mass, according to Chutkow et al., due to elevated levels of calorie intake and adipogenesis [67]. Despite having more fat, compared with the controls, Txnip knockout mice are significantly more insulin sensitive and exhibit enhanced glucose transport in both adipose and skeletal muscle tissues [67]. Txnip deficiency also directly affects PPAR expression and activity, implying Txnip is a novel mediator of insulin resistance and a regulator of adipogenesis. Txnip knockout mice are, thus, a promising mouse model for human MHO.

### 3.3. Tumor Progression Locus 2 (TPL2) Knockout Mice

Tumor progression locus 2 (TPL2) has been identified as an important modulator of immune responses that transmits inflammatory signals to downstream effectors, thereby modulating the generation and function of inflammatory cells [72]. TPL2 is activated by Toll-like receptor (TLR) ligands; cytokines, including tumor necrosis factor (TNF) family and interleukin (IL)-1β; and several chemokines [73,74]. Thus, knocking out TPL2 prevents cytokines (TNF- and IL-1) and proinflammatory stimuli (via lipopolysaccharide) from activating ERK and JNK [75,76]. In this context, TPL2 is in a unique position to integrate various inflammatory signaling pathways involved in the development and progression of obesity-related complications [68]. A study using a diet-induced obesity model with or without TPL2 knockout demonstrated that TPL2 deletion reduces peripheral inflammation and hepatic steatosis, and improves whole-body insulin resistance in obese mice, mimicking MHO observed in humans [68].

### 3.4. COL6 Knockout Mice

As previously noted, recent research suggests that excessive collagen and fibrosis exacerbate inflammatory and metabolic pathologies in obese patients. Particularly, collagen VI (COL6) is a highly enriched extracellular matrix component in adipose tissue [69]. Tayeba Khan et al. showed that obese (ob/ob) mice with COL6 knockdown, compared with ob/ob mice with intact COL6 expression, resulted in the uninhibited expansion of individual adipocytes but was paradoxically associated with substantial improvements in energy homeostasis such as better glucose tolerance and lower levels of circulating triacylglycerol after lipid challenge [69]. These findings indicate a possible role of COL6 in modulating adipocyte physiology and suggest COL6-KO mice as a potential murine model of human MHO.

### 3.5. Adipose-Specific GLUT4 Overexpression (AG4OX) Mice

GLUT4, the major insulin-responsive glucose transporter, plays a key role in systemic glucose metabolism in adipose tissue [77,78,79]. In insulin-resistant conditions, GLUT4 is downregulated in adipose tissue but not in muscle, the primary site of insulin-stimulated glucose uptake [77]. Moreover, mice with adipose-specific GLUT4 overexpression (AG4OX) have improved glucose homeostasis [79] while mice with adipose-specific GLUT4 knockout (AG4KO) have insulin resistance and type 2 diabetes [78]. Herman et al. revealed that AG4OX animals are more obese and insulin-sensitive than wild-type mice, which is consistent with the MHO phenotype [70]. In their study, the authors further showed that ChREBP, a glucose-responsive transcription factor that regulates fatty acid synthesis and glycolysis [80], is highly regulated by GLUT4 in adipose tissue and is a key determinant of systemic insulin sensitivity and glucose homeostasis, indicating that adipose ChREBP may be a novel strategy for preventing and treating obesity-related metabolic dysfunction [70].

### 3.6. MitoNEET Overexpression Mice

MitoNEET has been identified as a distinct dimeric mitochondrial membrane target that is crosslinked to pioglitazone [81,82]. MitoNEET was named based on its C-terminal amino acid sequence, AsnGluGluThr (NEET), which is found in the outer mitochondrial membrane [81]. MitoNEET achieves its effects on cellular and systemic metabolic homeostasis by acting as a potent iron content regulator in mitochondria. Kusminski et al. [71] created an adipose-specific transgenic model, an inducible tissue-specific overexpression system, and an inducible constitutive mitoNEET knockdown. The overexpression of mitoNEET disrupted the cellular energy balance by impairing mitochondrial activity, resulting in a decrease in oxidation and a compensatory increase in the cellular energy balance. This resulted in persistent adipose tissue development, and the mice in this model became extremely obese. Despite their obesity, mitoNEET overexpression during high caloric intake resulted in system-wide improvements in insulin sensitivity, providing a model of a metabolically healthy, obese state that minimizes lipotoxicity in tissues that are prone to storing lipids during excess caloric intake [71].

## 4. Definition and Concept of MHO

### 4.1. Clinical Definition of MHO

Currently, there is no internationally adopted standard for identifying MHO, and more than 30 distinct criteria have been employed to operationalize the symptoms in research [3,83]. Some criteria used to define metabolically healthy obesity are shown in Table 2. This may explain why the prevalence, stability, and clinical effects of MHO differ from study to study, contributing to an ongoing unresolved dispute [83]. Despite differences in definitions, some common elements of MHO are repeatedly empathized: healthy obesity denotes an absence of metabolic abnormalities in obese individuals, such as type 2 diabetes, dyslipidemia, and hypertension. When more data is available, measures of insulin resistance, such as homeostasis model assessment (HOMA) and inflammatory markers, are also utilized [4].

### 4.2. Dynamic Nature of MHO

Another barrier to determining the outcome of the MHO phenotype is its dynamic and changing nature. The health condition of a subject may change from metabolically healthy to metabolically unhealthy and vice versa. As a result, the clinical implications of MHO should be examined from the perspective of metabolic health being a transitory rather than permanent state. Approximately one-third to one-half of people with MHO progress to a metabolically unhealthy condition over time [88,89,90,91,92]. A healthier lifestyle; stronger incretin response to meals; less abdominal, visceral, and ectopic fat deposition; lower levels of inflammation; and insulin sensitivity are postulated attributes that help preserve a metabolically healthy state in individuals with MHO [4]. Maintaining these characteristics may therefore avoid the transition from MHO to MUO. These findings also imply that MHO is a dynamic condition that should be examined across time.

## 5. Clinical Outcomes of MHO and Possible Mechanisms

The predictive significance of MHO is a hotly debated topic that confronts significant challenges due to its varied definitions across studies and dynamic nature, as discussed above. In this context, our research team has investigated the effects of MHO, taking into account its phenotypic shift throughout time. In this section, we describe reported findings regarding the role of MHO in several outcomes, including mortality, cardiovascular risk, chronic kidney disease (CKD), dementia, and colorectal cancer. We also discuss some potential pathways for explaining the observed outcomes in patients with MHO (Table 3).

### 5.1. Mortality and Cardiovascular Event Risk

Obesity is a well-known risk factor for cardiovascular events (CVEs) and mortality. Although CVE risk is higher in patients with MHO than in metabolically healthy individuals with normal body weight, the risk is substantially lower in individuals with MHO, compared to those with MUO [95,96,102]. We analyzed the mortality and cardiovascular event rates in 514,866 participants from the Korean National Health Insurance Service–Health Examination Cohort [94] and found that the risk of CVE in the baseline MHO group was higher than that in the metabolically healthy nonobesity (MHNO) group (hazard ratio (HR), 1.14; 95% confidence interval (CI), 1.05–1.24). However, the all-cause mortality in the MHO group was lower than that in the MHNO group (HR, 0.86; 95% CI, 0.79–0.93). Among baseline MHO subjects, the risk of CVE was significantly higher in those who transitioned from MHO to MUO with a multivariate-adjusted HR of 1.24 (95% CI, 1.00–1.54), suggesting that weight loss and progression to a metabolically unhealthy state are the reasons behind the significant increase in mortality. 

The concept of the “obesity paradox” is rooted in the fact a higher BMI is associated with high incidence of type 2 diabetes, hypertension, dyslipidemia, and cardiovascular disease (CVD), obese individuals with these conditions may survive longer than leaner individuals [105,106]. Similarly, individuals classified as normal weight or underweight may have a poorer prognosis than overweight persons with respect to CVD, a condition termed the “lean paradox” [107]. Although the mechanism has not been fully elucidated, a progressive catabolic state and loss of lean muscle mass may result in improved outcomes for obese people and poorer ones for lean individuals [107]. Moreover, because obesity is a well-known cardiometabolic risk factor, more vigorous diagnostic testing and therapeutic interventions in the obese population may result in earlier testing and diagnosis, which may lead to better survival [107,108].

### 5.2. Chronic Kidney Disease

Obesity is a known risk factor for CKD and a serious public health issue globally [97,109,110]. Since obesity-induced metabolic disturbances, such as hypertension, insulin resistance, hyperglycemia, and dyslipidemia, are well-known factors in the development of CKD [111], the direct link between CKD and obesity or obesity-induced metabolic disturbances is unknown [4]. With regard to the risk of CKD in MHO subjects, prior studies have reported contradictory results [98,99,100,101]. In our longitudinal cohort study [112], the probability of incident CKD in the baseline MHO group was greater than that in the MHNO group (HR, 1.23; 95% CI, 1.12–1.36). Patients who converted to MHNO did not have an elevated risk (HR, 0.98; 95% CI, 0.72–1.32), whereas the stable MHO group and the groups that progressed to a metabolically unhealthy condition had a higher risk of incident CKD compared with the stable MHNO group. Although the processes by which obesity contributes to CKD are unknown, several possible explanations tying obesity to kidney disease that are independent of metabolic risk factors can be considered, including hemodynamic alterations, oxidative stress, and hormonal variables [113,114,115,116]. Obesity-induced renal impairment has been linked to changes in renal hemodynamics such as hyperfiltration, increased glomerular capillary wall tension, and podocyte stress [97,114]. Several adipokines, including leptin and adiponectin, and other adipose tissue-derived molecules, including TNF-α, IL-6, and plasminogen activator inhibitor-1, have been shown to impair renal function [115,117]. Although it is unclear whether these molecules have altered expression levels in MHO patients, these pathways may contribute to the development of incident CKD in obese individuals, particularly in those with MHO.

### 5.3. Dementia

Studies on the association between obesity and the development of Alzheimer’s disease (AD) indicate that midlife obesity is associated with a 1.7–2.0-fold increase in the risk of developing dementia and AD [118,119]. On the contrary, more recent publications report that being overweight or obese at old age protects against Alzheimer’s disease [120,121,122,123,124]. According to our nationwide population-based cohort study, the risk of AD is considerably reduced among people with MHO [125], which is consistent with some earlier findings [126,127]. In addition, we analyzed the risk of developing AD based on the changes in metabolic phenotype. Importantly, switching from MUO to MHO reduces the risk of AD development relative to maintaining a stable MHNO status (multivariable-adjusted HR, 0.62; 95% CI, 0.50–0.78), indicating a protective effect of MHO against AD. Several pathways are proposed to have a role in this protective effect. Insulin-like growth factor I, which has neurotrophic effects in the hippocampus [128,129,130,131], may play at least a partial role. Furthermore, adipokines released by adipose tissue, such as leptin, may also be involved [132]. Higher circulating leptin levels have been linked to a lower risk of dementia and Alzheimer’s disease, and increased brain volume [133,134,135]. As a result, altered levels of hormones and adipokines in patients with MHO may increase dementia risk, albeit further research is needed to better understand the underlying mechanisms.

### 5.4. Colorectal Cancer

Obesity is a well-known risk factor for CRC; however, only a few studies have investigated whether obese patients without metabolic abnormalities are at increased risk of CRC. A prospective cohort study in Korea showed that the MHO phenotype is a risk factor for CRC in men [104]. However, recently, Cao et al. used data derived from 390,575 adults from the UK Biobank and reported that even in metabolically healthy individuals, obesity was associated with increased risks of colorectal cancers [103]. Our study on a nationwide population-based cohort suggested that metabolic unhealthiness significantly contributes to incident CRC in the obese population [136]. The stable MHO group showed no increased risk of incident CRC (HR, 0.97; 95% CI, 0.83–1.14). However, the group transitioning from MHO to MUO had a higher risk of incident CRC compared with the stable MHNO group (HR, 1.34; 95% CI, 1.15–1.57). Among the patients with baseline MUO, those transitioning into MHO were not at increased risk of CRC (HR, 1.06; 95% CI, 0.91–1.25), whereas those who remained in the stable MUO group had a higher risk of incident CRC compared with those in the stable MHNO group (HR, 1.29; 95% CI, 1.19–1.41). Previously, Ko et al. reported that for advanced CRN, metabolically unhealthy status (i.e., MUNO or MUO) increased the risk but MHO did not [137]. However, MHO subjects were at an increased risk of CRN in general; based on these findings, the authors proposed that metabolically unhealthy status may be the step after simple obesity in the process of colorectal carcinogenesis via increased levels of growth factors (e.g., insulin-like growth factor or epidermal growth factor receptor) by insulin resistance, which leads to advanced cancer [137]. Therefore, chronic inflammation and disturbance of adipokines or growth factors in obesity could be potential mechanisms linking obesity and cancer, which was proposed from the studies on MHO subjects [136,137,138].

## 6. Summary and Conclusions

MHO is not an entirely new concept. Numerous possible mechanisms underlying MHO have been suggested, including adipose tissue distribution, inflammation and fibrosis in adipose tissue, and altered adipokine secretion. Murine models of metabolically protected obesity, with a salutary profile of adipose tissue function and energy metabolism, have provided robust insights into the human MHO phenotype. Clinically, the prognostic value of MHO is a subject of debate and the impact of MHO on obesity-related morbidity and mortality requires further investigation. Further efforts are needed to establish a unified definition of MHO to develop effective treatment strategies and to discover the pathophysiologic underpinnings of MHO.

## Figures and Tables

**Figure 1 ijms-23-09614-f001:**
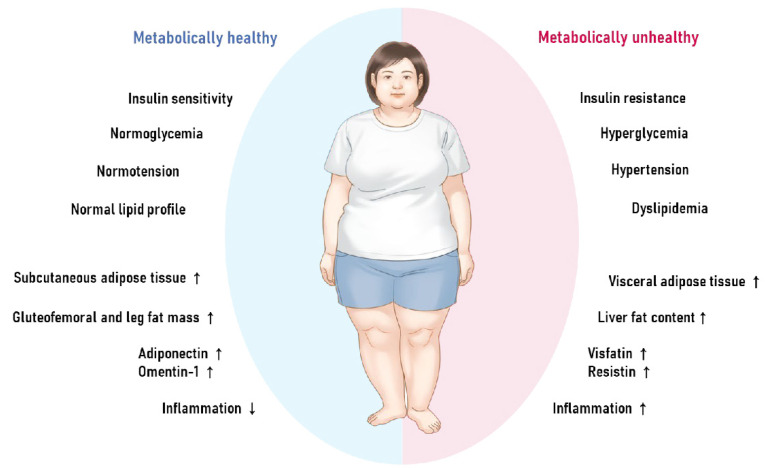
The characteristics and physiology of metabolically healthy obesity versus metabolically unhealthy obesity.

**Table 1 ijms-23-09614-t001:** Murine models representing the human metabolically healthy obesity phenotype.

Molecule	Background	Phenotype	Reference
Adiponectin transgenic mouse	Adiponectin is an anti-inflammatory, insulin-sensitizing adipokine expressed by adipocytes that improves lipid and glucose metabolism	Increased level of plasma adiponectin, lower inflammation, andbetter insulin sensitivity than ob/ob littermates	[66]
Txnip knockout mice	Thioredoxin-interacting protein (Txnip) is a cellular oxidative stress regulator, which limits glucose absorption into fat and muscle	Increased adipogenesis, preserved insulin sensitivity, increased glucose transport to adipose tissue and muscle	[67]
TPL2 knockout mice	Tumor progression locus 2 (TPL2) is a serine/threonine kinase that functions downstream of IKK-β and integrates inflammatory signaling pathways	Reduced inflammation and hepatic steatosis, improved insulin resistance	[68]
COL6 knockout mice	Collagen VI is a highly enriched extracellular matrix component of adipose tissue	Increased amount of adipose tissue, lower fasting glucose and improved glucose tolerance	[69]
Adipose-specific GLUT4 over-expression mice	GLUT4, the major insulin-responsive glucose transporter, plays a key role in systemic glucose metabolism in adipose tissue	More obese and insulin-sensitive than wild-type mice	[70]
MitoNEET overexpression mice	MitoNEET has been identified as a distinct dimeric mitochondrial membrane target that is crosslinked to pioglitazone	Extremely obese but improved insulin sensitivity during high caloric intake	[71]

**Table 2 ijms-23-09614-t002:** Various criteria used to define metabolically healthy obesity in clinical investigations.

Variable/Study	NECP-ATP III (2001) [84]	Karelis et al. (2004) [85]	Wildman et al. (2008) [86]	Stefan et al. (2008) [20]	Aguilar-Salinas et al. (2008) [25]	Zembic et al. (2021) [87]
Metabolic components						
BP, mmHg	≥130/85 or treatment		≥130/85 or treatment		<140/90 and no treatment	Systolic BP ≥130 or treatment
FPG, mg/dL	≥100 or treatment		≥100 or treatment		<126 and no treatment	Prevalent diabetes
TG, mg/dL	≥150	<150	≥150			
HDL, mg/dL	<40 (men) <50 (women)	≥50	<40 (men) <50 (women)		≥40	
HOMA-IR		<1.95	>90th percentile			
WC, cm	≥102 (men)≥88 (women)					
Others		TC <200 mg/dL LDL <100 mg/dL	hsCRP >90th percentile	WBISI >75th percentile		WHR ≥1.03 (men) ≥0.95 (women)
Metabolic health criteria	<3 of the above	≥4 of the above	<2 of the above	All of the above	All of the above	None of the above
Obesity components						
BMI, kg/m^2^	≥25	≥30	≥30	≥30	≥25	≥30

NECP-ATP III, National Cholesterol Education Program Adult Treatment Panel III; WC, waist circumference; BP, blood pressure; FPG, fasting plasma glucose; TG, triglyceride; HDL, high density lipoprotein; HOMA-IR, homeostasis model assessment of insulin resistance; TC, total cholesterol; LDL, low density lipoprotein; hsCRP, high-sensitivity C-reactive protein; WBISI, whole body insulin sensitivity index; WHR; waist-to-hip ratio; BMI, body mass index.

**Table 3 ijms-23-09614-t003:** Clinical outcomes of metabolically healthy obesity versus unhealthy obesity.

Outcome	HR (95% CI) for MHO (with MHNO as the Reference)	HR (95% CI) for MUO (with MHNO as the Reference)	Reference
Mortality	1.81 (1.16–2.84)	2.01 (1.43–2.83)	[93]
	0.86 (0.79–0.93)	0.96 (0.91–1.01)	[94]
	0.98 (0.87–1.10)	1.24 (1.16–1.32)	[87]
Cardiovascular events	1.45 (1.20–1.70)	2.31 (1.99–2.69)	[95]
	1.49 (1.45–1.54)	2.05 (1.94–2.16) ^1^ 2.41 (2.25–2.58) ^2^ 2.91 (2.68–3.18) ^3^	[96]
	1.14 (1.05–1.24)	1.55 (1.47–1.63)	[94]
Chronic kidney disease	0.83 (0.36–1.70)	2.80 (1.45–5.35)	[97]
	1.23 (1.12–1.36)	1.98 (1.85–2.10)	[83]
	1.17 (0.89–1.53) 2.21 (1.59–3.08) 2.20 (1.55–3.11)	2.25 (1.91–2.65) ^4^ 2.75 (2.32–3.25) ^5^ 4.02 (3.40–4.75) ^6^	[98] ^7^
	1.52 (0.93–2.49)	2.20 (1.44–3.38)	[99]
	0.95 (0.49–1.83) (men) 0.95 (0.74–1.20) (women)	2.22 (1.44–3.41) (men) 1.23 (1.01–1.54) (women)	[100]
Alzheimer’s disease	0.73 (0.54–0.97)	0.93 (0.70–1.24)	[101]
	0.73 (0.65-0.81)	0.96 (0.90-1.03)	[96]
Colorectal cancer	1.14 (1.04–1.26)	1.21 (1.13–1.29)	[102]
	1.10 (0.92–1.33)	1.29 (1.14–1.47)	[103]
	1.21 (1.06–1.39) (men) 1.10 (0.94–1.28) (women)	1.32 (1.19–1.48) (men) 1.08 (0.95–1.23) (women)	[104]

CI, confidence interval; HR, hazard ratio; MHNO, metabolically healthy nonobesity; MHO, metabolically healthy obesity; MUO, metabolically unhealthy obesity. ^1^ Obesity with 1 metabolic risk factor; ^2^ Obesity with 2 metabolic risk factors; ^3^ Obesity with 3 metabolic risk factors; ^4^ class I obesity, body mass index (BMI) 30–34.9 kg/m^2^; ^5^ class II obesity, BMI 35–39.9 kg/m^2^; ^6^ class III obesity, BMI ≥40 kg/m^2^; ^7^ HR for kidney function decline defined as eGFR decline ≥ 30%.

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
