# Peer review of "Pathogenesis, Murine Models, and Clinical Implications of Metabolically Healthy Obesity"

_ijms, 2022, doi:10.3390/ijms23179614_

Round 1
Reviewer 1 Report
In this review article the authors describe the pathophysiology and some murine models of MHO. The definition, variability, and clinical implications of the MHO phenotype is also discussed. The authors concluded that understanding the characteristics which differentiate people with MHO from those with MUO can lead to new insights into the mechanisms behind obesity-related metabolic derangements and diseases.
Specific comments:
1. Introduction: When the authors discuss the health burden of obesity, they should address the findings of more recent publications (N Engl J Med. 2017 Jul 6;377(1):13-27.; PLoS Med. 2020 Jul 28;17(7):e1003198.) that carefully evaluated these relationships, than of publications from 2003.
2. Body composition and fat distribution: It has now been established that precisely measured upper- and lower-body fat mass independently and differently affect the cardiometabolic risk). In this respect the authors should also carefully address the role of abdominal and gluteofemoral adipose tissue, as was discussed in Lancet Diabetes Endocrinol. 2020 Jul;8(7):616-627. The respective information also needs to be highlighted in the figure 1.
3. Body composition and fat distribution: It has also been well established that fatty liver strongly separates MUHO from MHO (Arch Intern Med. 2008 Aug 11;168(15):1609-16; Lancet Diabetes Endocrinol. 2020 Jul;8(7):616-627). The authors should carefully discuss this aspect and highlight it in the figure 1. In addition, the authors need to address that, while fatty liver stronger than visceral obesity separates MUHO from MHO, there is some variability in the pathogenesis of fatty liver and most of the genetically-determined hepatic steatosis is not associated with an increased cardiometabolic risk (Lancet Diabetes Endocrinol. 2022 Apr;10(4):284-296).
4. Papers supporting the statement ‘markers such as ghrelin and PAI-1 are solely associated with MUO, not with MHO’ should be cited.
5. When the authors address murine models representing human metabolically healthy obesity phenotype, they should also discuss the very interesting mouse models with overexpression of adipose-specific GLUT4 (Nature. 2012 Apr 19;484(7394):333-8) and mitoNEET (Nat Med. 2012 Oct;18(10):1539-49.).
Author Response
In this review article the authors describe the pathophysiology and some murine models of MHO. The definition, variability, and clinical implications of the MHO phenotype is also discussed. The authors concluded that understanding the characteristics which differentiate people with MHO from those with MUO can lead to new insights into the mechanisms behind obesity-related metabolic derangements and diseases.
Specific comments:
- Introduction: When the authors discuss the health burden of obesity, they should address the findings of more recent publications (N Engl J Med. 2017 Jul 6;377(1):13-27.; PLoS Med. 2020 Jul 28;17(7):e1003198.) that carefully evaluated these relationships, than of publications from 2003.
Response: Thank you for this comment. As per your suggestion, we referred to more updated publications which you recommended (N Engl J Med. 2017 Jul 6;377(1):13-27.; PLoS Med. 2020 Jul 28;17(7):e1003198.), in our revised manuscript as follows:
“Obesity is related with a decreased life expectancy, mostly because obese individuals have a higher risk of developing type 2 diabetes, cardiovascular disease (CVD), and cancer [1, 2].” (line 30 - line 32 in the revised manuscript)
- Body composition and fat distribution: It has now been established that precisely measured upper- and lower-body fat mass independently and differently affect the cardiometabolic risk. In this respect the authors should also carefully address the role of abdominal and gluteofemoral adipose tissue, as was discussed in Lancet Diabetes Endocrinol. 2020 Jul;8(7):616-627. The respective information also needs to be highlighted in the figure 1.
Response: We sincerely appreciate this insightful comment. Based on your comment and the recommended article (Lancet Diabetes Endocrinol. 2020 Jul;8(7):616-627), we discussed the metabolically unhealthy fat distribution, such as a lower percentage of gluteofemoral fat mass and an increased visceral fat mass, which are significantly correlated with cardiometabolic risk. In this regard, we discussed this issue in our revised manuscript and revised Figure 1.
"According to several studies, it is indisputable that increasing visceral adiposity independently correlates with a higher cardiometabolic risk [14-19]. In contrast, the metabolic profile of subcutaneous fat is more favorable [14, 20]. Using whole-body MRI and spectroscopy, the German Tübingen diabetes family study evaluated total, visceral, subcutaneous abdominal, gluteofemoral and leg fat mass, and liver fat content to distinguish between individuals with healthy and unhealthy obesity [20]. In this study, a lower proportion of gluteofemoral fat mass and leg fat mass was found to be the most reliable predictor for a metabolically unhealthy condition. Increased glu-teofemoral and leg fat mass, as well as higher insulin sensitivity and preserved insulin secretion, determined a metabolically healthy status, while increased visceral fat mass, increased subcutaneous abdominal fat mass, and high liver fat content contributed to a metabolically unhealthy phenotype [20, 21].” (line 71 - line 82 in the revised manuscript)
“Collectively, assessment of body fat distribution may be the key to understanding the pathophysiology linking obesity, metabolic health and cardiometabolic risk.” (line 86 - line 88 in the revised manuscript)
- Body composition and fat distribution: It has also been well established that fatty liver strongly separates MUHO from MHO (Arch Intern Med. 2008 Aug 11;168(15):1609-16; Lancet Diabetes Endocrinol. 2020 Jul;8(7):616-627). The authors should carefully discuss this aspect and highlight it in the Figure 1. In addition, the authors need to address that, while fatty liver stronger than visceral obesity separates MUHO from MHO, there is some variability in the pathogenesis of fatty liver and most of the genetically-determined hepatic steatosis is not associated with an increased cardiometabolic risk (Lancet Diabetes Endocrinol. 2022 Apr;10(4):284-296).
Response: I appreciate your informative comment. As you commented, a fatty liver is one of the most significant predictors of metabolic unhealthiness and one of its important characteristics. As stated in our amended text in response to your previous comment, we discussed this issue as follows:
“Using whole-body MRI and spectroscopy, the German Tübingen diabetes family study evaluated total, visceral, subcutaneous abdominal, gluteofemoral and leg fat mass, and liver fat content to distinguish between individuals with healthy and unhealthy obesity [20].” (line 73 - line 76 in the revised manuscript)
“Increased gluteofemoral and leg fat mass, as well as higher insulin sensitivity and pre-served insulin secretion, determined a metabolically healthy status, while increased visceral fat mass, increased subcutaneous abdominal fat mass, and high liver fat content contributed to a metabolically unhealthy phenotype [20, 21]. Another German study with 314 white people found that the proportion of liver fat is a significant pre-dictor of metabolically benign obesity [20]. Non-alcoholic liver disease (NAFLD) is strongly associated with the metabolically unhealthy obesity phenotype, although some subtypes of NAFLD with a stronger hepatic genetic component are not associated with insulin resistance and cardiometabolic risk [22, 23].” (line 78 - line 86 in the revised manuscript)
- Papers supporting the statement ‘markers such as ghrelin and PAI-1 are solely associated with MUO, not with MHO’ should be cited.
Response: Thank you for this comment. We indicated the citation (Life (Basel). 2021 Dec; 11(12): 1350) for the statement in our revised manuscript.
“Furthermore, markers such as ghrelin and PAI-1 are solely associated with MUO, not with MHO [36].” (line 108 - line 109 in the revised manuscript)
- When the authors address murine models representing human metabolically healthy obesity phenotype, they should also discuss the very interesting mouse models with overexpression of adipose-specific GLUT4 (Nature. 2012 Apr 19;484(7394):333-8) and mitoNEET (Nat Med. 2012 Oct;18(10):1539-49.).
Response: Thank you for letting us know the interesting mouse models. In response to your comment, the following was added to our discussion: adipose-specific GLUT4 overexpression (AG4OX) mice and mitoNEET mice. We also added those two models in our Table 1.
“3.5. Adipose-specific GLUT4 overexpression (AG4OX) mice
GLUT4, the major insulin-responsive glucose transporter, plays a key role in sys-temic glucose metabolism in adipose tissue [77-79]. In insulin-resistant conditions, GLUT4 is downregulated in adipose tissue but not in muscle, the primary site of insu-lin-stimulated glucose uptake [77]. Moreover, mice with adipose-specific GLUT4 overexpression (AG4OX) have improved glucose homeostasis [79], while mice with adipose-specific GLUT4 knockout (AG4KO) have insulin resistance and type 2 diabetes [78]. Herman et al. revealed that AG4OX animals are more obese and insulin-sensitive than wild-type mice, which is consistent with the MHO phenotype [70]. In their study, the authors further showed that ChREBP, a glucose-responsive transcription factor that regulates fatty acid synthesis and glycolysis [80], is highly regulated by GLUT4 in adipose tissue and is a key determinant of systemic insulin sensitivity and glucose ho-meostasis, indicating that adipose ChREBP may be a novel strategy for preventing and treating obesity-related metabolic dysfunction [70].” (line 192 - line 205 in the revised manuscript)
”3.6. MitoNEET overexpression mice
MitoNEET has been identified as a distinct dimeric mitochondrial membrane tar-get that is crosslinked to pioglitazone [81, 82]. MitoNEET was called based on its C-terminal amino acid sequence, AsnGluGluThr (NEET), which is found in the outer mitochondrial membrane [81]. MitoNEET achieves its effects on cellular and systemic metabolic homeostasis by acting as a potent iron content regula-tor in mitochondria. Kusminski et al. [71] created an adipose-specific transgenic model, an inducible tis-sue-specific overexpression system, and an inducible constitu-tive mitoNEET knock-down. The overexpression of mitoNEET disrupted the cellular energy balance by im-pairing mitochondrial activity, resulting in a decrease in -oxidation and a compensa-tory increase in cellular energy balance. This resulted in persistent adipose tissue de-velopment, and the mice in this model got extremely obese. Despite their obesity, mi-toNEET overexpression during high caloric intake resulted in system-wide improve-ments in insulin sensitivity, providing a model of a metabolically healthy, obese state that minimizes lipotoxicity in tissues that are prone to store lipids during excess caloric intake [71].” (line 207 - line 221 in the revised manuscript)
Table 1. Murine models representing human metabolically healthy obesity phenotype.
|
Molecule |
Background |
Phenotype |
Reference |
|
Adiponectin |
Adiponectin is an anti-inflammatory, insulin-sen sitizing adipokine expressed by adipocytes that improves lipid and glucose metabolism |
Increased level of plasma adiponec tin, lower inflammation, better insulin sensitivity than ob/ob littermates |
[66] |
|
Txnip |
Thioredoxin interacting protein (Txnip) is a cellu lar oxidative stress regulator which limits glucose absorption into fat and muscle |
Increased adipogenesis, preserved in sulin sensitivity, increased glucose transport to adipose tissue and muscle |
[67] |
|
TPL2 |
Tumor progression locus 2 (TPL2) is a serine/thre onine kinase that functions downstream of IKK-β and integrates inflammatory signaling pathways |
Reduced inflammation and hepatic steatosis, improved insulin resistance |
[68] |
|
COL6 |
Collagen VI is a highly enriched extracellular ma trix component of adipose tissue |
Increased amount of adipose tissue, lower fasting glucose and improved glucose tolerance |
[69] |
|
Adipose-specific GLUT4 over- |
GLUT4, the major insulin-responsive glucose transporter, plays a key role in systemic glucose metabolism in adipose tissue |
More obese and insulin-sensitive than wild-type mice |
[70] |
|
MitoNEET |
MitoNEET has been identified as a distinct di meric mitochondrial membrane target that is crosslinked to pioglitazone |
Extremely obese but improved insu lin sensitivity during high caloric intake |
[71] |

Reviewer 2 Report
This manuscript has provided a concise review of the phenotype of MHO/MUO, the possible contributing mechanisms/factors, definition of MHO and the long term clinical outcomes of MHO. In particular, the review of the animal models of MHO is very important and informative. Minor revisions are required for the manuscript.
Title: The title “Metabolically healthy obesity: from bench to clinic” does not seem appropriate for the review as no in-depth bench work/in vitro experiments were described in this review. Please kindly revise the title.
Section 1. Introduction:
Line 37: Please insert “associated” and revise to “commonly associated with chronic inflammatory state”.
Line 39: Since metabolically healthy obesity is distinguished by absence of metabolic abnormalities, what do the authors mean by “some animal models of MHO based on these pathophysiologic abnormalities”? MHO should not have abnormalities based on the authors’ earlier definition.
Line 41: Please change “instable” to “unstable”.
Section 3.1
Line 126: The authors mentioned that the hyper obese mice overexpressing adiponectin have fewer adipocytes but it was stated in line 122 that these mice have more adipose tissue. Please clarify.
Section 4.1: The authors should provide some examples of the different MHO definitions used and cite the appropriate references.
Section 4.2:
Line 178: Please remove “,”.
Line 183: Please change “MHU” to “MHO”.
Section 5:
Line 189: Please change “our” to “reported”.
Table 2 legend: Please remove MUNO since MUNO is not mentioned in Table 2.
Table 2: Can the authors please explain how did they derive the HR? For example, reference 74 is on the incidence/risk of T2DM in MHO subjects, how was the HR for mortality derived?
Lines 204: The authors stated that “We analyzed the mortality and cardiovascular” so does that mean the authors reanalysed the data in terms of mortality and CVE?
Line 212: Why is weight loss behind the significant increase in mortality?
Section 6:
Line 283-285: Please change to “Clinically, the prognostic value of MHO is a subject of debate and the impact of MHO on obesity-related morbidity and mortality requires further investigation”.
Lines 281-288: The authors could include the importance of using MHO animal models to further understand the underlying mechanisms contributing to the pathophysiology of MHO.
Author Response
Reviewer #2.
This manuscript has provided a concise review of the phenotype of MHO/MUO, the possible contributing mechanisms/factors, definition of MHO and the long term clinical outcomes of MHO. In particular, the review of the animal models of MHO is very important and informative. Minor revisions are required for the manuscript.
Title: The title “Metabolically healthy obesity: from bench to clinic” does not seem appropriate for the review as no in-depth bench work/in vitro experiments were described in this review. Please kindly revise the title.
Response: We sincerely appreciate this insightful comment. According to your suggestion, we revised the title of our review as “Pathogenesis, Murine Models, and Clinical Implications of Metabolically Healthy Obesity”.
Section 1. Introduction:
Line 37: Please insert “associated” and revise to “commonly associated with chronic inflammatory state”.
Response: Thank you for this comment. We revised the sentence as follows:
“MHO is distinguished by the absence of metabolic disturbances, including insulin resistance, type 2 diabetes, hypertension and dyslipidemia, commonly associated with chronic inflammatory state [4, 5].” (line 37 - line 39 in the revised manuscript)
Line 39: Since metabolically healthy obesity is distinguished by absence of metabolic abnormalities, what do the authors mean by “some animal models of MHO based on these pathophysiologic abnormalities”? MHO should not have abnormalities based on the authors’ earlier definition.
Response: Thank you for this comment. We apologize for our imprecise sentence. We clarified the phrase as follows:
“In this paper, we compare the pathophysiology of MHO with that of metabolically unhealthy obesity (MUO), and then describe some animal models of MHO based on the physiology and features of this phenotype.” (line 39 - line 42 in the revised manuscript)
Line 41: Please change “instable” to “unstable”.
Response: Thank you for this comment. As per your suggestion, we changed change “instable” to “unstable” as follows:
“We will also address the concept of MHO and its clinical consequences given the un-stable nature of this phenotype.” (line 42 - line 43 in the revised manuscript)
Section 3.1
Line 126: The authors mentioned that the hyper obese mice overexpressing adiponectin have fewer adipocytes but it was stated in line 122 that these mice have more adipose tissue. Please clarify.
Response: We apologize our carelessness. We meant that the hyper-obese mice overexpressing adiponectin had smaller (not fewer) adipocytes though the number of adipocytes increased. To clarify this, we revised the sentence as below:
“Overexpression of adiponectin results in the development of hyper-obese animals that exhibit subcutaneous fat as the most abundant type of adipose tissue, a larger number of adipocytes with much smaller average cell size, reduced inflammation, and metabolic fitness, all of which are specific features of MHO [66].” (line 150 - line 153 in the revised manuscript)
Section 4.1: The authors should provide some examples of the different MHO definitions used and cite the appropriate references.
Response: We sincerely appreciate this comment. As per your suggestion, we provided some examples of the various MHO definitions in our revised manuscript (in the text and Table 2), as follows:
“Currently, there is no internationally adopted standard for identifying MHO, and more than 30 distinct criteria have been employed to operationalize the symptoms in research [3, 83]. Some criteria used to define metabolically healthy obesity is shown in Table 2.” (line 225 - line 227 in the revised manuscript)
Table 2. Various criteria used to define metabolically healthy obesity in clinical investigations.
|
Variable/study |
NECP-ATP III (2001) [84] |
Karelis et al. |
Wildman et al. (2008) [86] |
Stefan et al. (2008) [20] |
Aguilar-Salinas et al. (2008) [25] |
Zembic et al. (2021) [87] |
|
Metabolic |
|
|
|
|
|
|
|
BP, mmHg |
≥ 130/85 or treatment |
|
≥ 130/85 or treatment |
|
< 140/90 and no treatment |
Systolic BP ≥ 130 or treatment |
|
FPG, mg/dL |
≥ 100 or treatment |
|
≥ 100 or treatment |
|
< 126 and no treatment |
Prevalent diabetes |
|
TG, mg/dL |
≥ 150 |
< 150 |
≥ 150 |
|
|
|
|
HDL, mg/dL |
< 40 (men) < 50 (women) |
≥ 50 |
< 40 (men) < 50 (women) |
|
≥ 40 |
|
|
HOMA-IR |
|
< 1.95 |
> 90th percentile |
|
|
|
|
WC, cm |
≥ 102 (men) |
|
|
|
|
|
|
Others |
|
TC < 200 mg/dL LDL < 100 mg/dL |
hsCRP > 90th percentile |
WBISI > 75th percentile |
|
WHR |
|
Metabolic health criteria |
< 3 of the above |
≥ 4 of the above |
< 2 of the above |
All of the above |
All of the above |
None of the above |
|
Obesity |
|
|
|
|
|
|
|
BMI, kg/m2 |
≥ 25 |
≥ 30 |
≥ 30 |
≥ 30 |
≥ 25 |
≥ 30 |
NECP-ATP III, National Cholesterol Education Program Adult Treatment Panel III; WC, waist circumference; BP, blood pressure; FPG, fasting plasma glucose; TG, triglyceride; HDL, high density lipoprotein; HOMA-IR, homeostasis model assessment of insulin resistance; TC, total cholesterol; LDL, low density lipoprotein; hsCRP, high-sensitivity C-reactive protein; WBISI, whole body insulin sensitivity index; WHR; waist-to-hip ratio; BMI, body mass index.
Section 4.2:
Line 178: Please remove “,”.
Response: Thank you for this comment. We removed “,” in the sentence as follows:
“As a result, the clinical implications of MHO should be examined in the perspective of metabolic health being a transitory, rather than permanent state.” (line 245 - line 247 in the revised manuscript)
Line 183: Please change “MHU” to “MHO”.
Response: We apologize our carelessness. We corrected the typo as follows:
“Maintaining these characteristics may therefore avoid the transition from MHO to MUO.” (line 251 - line 252 in the revised manuscript)
Section 5:
Line 189: Please change “our” to “reported”.
Response: Thank you for this comment. We rewrote the sentence as follows:
“In this section, we describe reported findings regarding the role of MHO in several outcomes, including mortality, cardiovascular risk, chronic kidney disease (CKD), dementia, and colorectal cancer.” (line 258 - line 260 in the revised manuscript)
Table 2 legend: Please remove MUNO since MUNO is not mentioned in Table 2.
Response: Thank you for this comment. We removed MUNO in Table 2 (Table 3 in the revised version) legend.
Table 2: Can the authors please explain how did they derive the HR? For example, reference 74 is on the incidence/risk of T2DM in MHO subjects, how was the HR for mortality derived?
Response: Thank you for this comment. Reference 74 (Reference 94 in the revised manuscript) is on the risk of cardiovascular events and mortality according to the metabolic health and obesity status, not on the incidence/risk of T2DM (Title: Implications of the dynamic nature of metabolic health status and obesity on risk of incident cardiovascular events and mortality: a nationwide population-based cohort study. Metabolism 2019, 97, 50-56.). In this study, we categorized study participants into four groups (i.e. MHNO, MHO, MUNO, and MUO), and calculated HRs for cardiovascular events, CV mortality and all-cause mortality, using MHNO group as the reference.
Lines 204: The authors stated that “We analyzed the mortality and cardiovascular” so does that mean the authors reanalysed the data in terms of mortality and CVE?
Response: As we described above, Reference 74 (Reference 94 in the revised manuscript) is on the risk of cardiovascular events and mortality according to the metabolic health and obesity status. In this study, we categorized study participants into four groups (i.e. MHNO, MHO, MUNO, and MUO), and calculated HRs for cardiovascular events, CV mortality and all-cause mortality, using MHNO group as the reference.
Line 212: Why is weight loss behind the significant increase in mortality?
Response: We sincerely appreciate this insightful comment. As widely known, The term “obesity paradox” was introduced as a concept explaining that, although higher BMI is related to increased rates of hypertension, dyslipidemia, type 2 diabetes, and CVD, obese individuals with these conditions may have better outcomes than leaner individuals (Cardiol Rev. 2014 Jul-Aug;22(4):163-70, Obes Rev. 2018 Sep;19(9):1236-1247). Similarly, individuals classified as normal weight or underweight may have a poorer prognosis than overweight persons with respect to CVD, a condition termed the “lean paradox” (Prog Cardiovasc Dis. 2018 Jul-Aug;61(2):142-150). Better outcomes in obese populations, or poorer prognoses in lean subjects, may result from a progressive catabolic state and loss of lean mass. Additionally, as obesity is well known as a cardiometabolic risk factor, higher pretest probability for CVD and earlier diagnostic testing in obese individuals could lead to earlier testing and earlier diagnosis, which could result in increased survival (Prog Cardiovasc Dis. 2018 Jul-Aug;61(2):142-150, Prog Cardiovasc Dis. 2014 Jan-Feb;56(4):401-8). We discussed this issue in more detail in our revised manuscript as follows:
“The concept of “obesity paradox” is rooted in the fact a higher BMI is associated with high incidence of type 2 diabetes, hypertension, dyslipidemia, and cardiovascular disease (CVD), obese individuals with these conditions may survive longer than leaner individuals [108, 109]. Similarly, individuals classified as normal weight or under-weight may have a poorer prognosis than overweight persons with respect to CVD, a condition termed the “lean paradox” [110]. Although the mechanism has not been fully elucidated, a progressive catabolic state and loss of lean muscle mass may result in improved outcomes for obese people and poorer ones for lean individuals [110]. Moreover, because obesity is a well-known cardiometabolic risk factor, more vigorous diagnostic testing and therapeutic interventions in obese population may result in ear-lier testing and diagnosis, which may lead to better survival [110, 111].” (line 286 - line 296 in the revised manuscript)
Section 6:
Line 283-285: Please change to “Clinically, the prognostic value of MHO is a subject of debate and the impact of MHO on obesity-related morbidity and mortality requires further investigation”.
Response: Thank you for this instructive comment. We revised the sentence as per your suggestion.
“Clinically, the prognostic value of MHO is a subject of debate and the impact of MHO on obesity-related morbidity and mortality requires further investigation.” (line 369 - line 371 in the revised manuscript)
Lines 281-288: The authors could include the importance of using MHO animal models to further understand the underlying mechanisms contributing to the pathophysiology of MHO.
Response: We sincerely appreciate this insightful comment. As per your suggestion, we included discussion on the importance of murine models of MHO phenotype as follows:
“Murine models of metabolically protected obesity, with a salutary profile of adipose tissue function and energy metabolism, have provided robust insights into human MHO phenotype.” (line 367 - line 369 in the revised manuscript)

Reviewer 3 Report
The authors provide a short but insightful view on the definition, models and health outcomes of metabolically healthy obesity.
This is not a novel topic, yet it is interesting and relevant for the field. The review is well written and organized. I believe that one important mouse model that has to be added to this review to make it more complete and comprehensive is the ChREBPβ found in the paper "A novel ChREBP isoform in adipose tissue regulates systemic glucose metabolism" published in Nature 2012.
I would also encourage the authors to refer to a broader body of work and not restrict some of the sections to their observations/previous publications. For example 5.2. Chronic kidney disease and 5.4. Colorectal cancer
In general the quality of this review is good but would benefit a lot by the suggestions above.
Author Response
Reviewer #3.
The authors provide a short but insightful view on the definition, models and health outcomes of metabolically healthy obesity.
This is not a novel topic, yet it is interesting and relevant for the field. The review is well written and organized. I believe that one important mouse model that has to be added to this review to make it more complete and comprehensive is the ChREBPβ found in the paper "A novel ChREBP isoform in adipose tissue regulates systemic glucose metabolism" published in Nature 2012.
Response: We sincerely appreciate this insightful comment. We included the important murine model you commented (Nature. 2012 Apr 19;484(7394):333-8.) in our revised manuscript as follows:
“3.5. Adipose-specific GLUT4 overexpression (AG4OX) mice
GLUT4, the major insulin-responsive glucose transporter, plays a key role in sys-temic glucose metabolism in adipose tissue [77-79]. In insulin-resistant conditions, GLUT4 is downregulated in adipose tissue but not in muscle, the primary site of insu-lin-stimulated glucose uptake [77]. Moreover, mice with adipose-specific GLUT4 overexpression (AG4OX) have improved glucose homeostasis [79], while mice with adipose-specific GLUT4 knockout (AG4KO) have insulin resistance and type 2 diabetes [78]. Herman et al. revealed that AG4OX animals are more obese and insulin-sensitive than wild-type mice, which is consistent with the MHO phenotype [70]. In their study, the authors further showed that ChREBP, a glucose-responsive transcription factor that regulates fatty acid synthesis and glycolysis [80], is highly regulated by GLUT4 in adipose tissue and is a key determinant of systemic insulin sensitivity and glucose ho-meostasis, indicating that adipose ChREBP may be a novel strategy for preventing and treating obesity-related metabolic dysfunction [70].” (line 192 - line 205 in the revised manuscript)
Table 1. Murine models representing human metabolically healthy obesity phenotype.
|
Molecule |
Background |
Phenotype |
Reference |
|
Adiponectin |
Adiponectin is an anti-inflammatory, insulin-sen sitizing adipokine expressed by adipocytes that improves lipid and glucose metabolism |
Increased level of plasma adiponec tin, lower inflammation, better insulin sensitivity than ob/ob littermates |
[66] |
|
Txnip |
Thioredoxin interacting protein (Txnip) is a cellu lar oxidative stress regulator which limits glucose absorption into fat and muscle |
Increased adipogenesis, preserved in sulin sensitivity, increased glucose transport to adipose tissue and muscle |
[67] |
|
TPL2 |
Tumor progression locus 2 (TPL2) is a serine/thre onine kinase that functions downstream of IKK-β and integrates inflammatory signaling pathways |
Reduced inflammation and hepatic steatosis, improved insulin resistance |
[68] |
|
COL6 |
Collagen VI is a highly enriched extracellular ma trix component of adipose tissue |
Increased amount of adipose tissue, lower fasting glucose and improved glucose tolerance |
[69] |
|
Adipose-specific GLUT4 over- |
GLUT4, the major insulin-responsive glucose transporter, plays a key role in systemic glucose metabolism in adipose tissue |
More obese and insulin-sensitive than wild-type mice |
[70] |
|
MitoNEET |
MitoNEET has been identified as a distinct di meric mitochondrial membrane target that is crosslinked to pioglitazone |
Extremely obese but improved insu lin sensitivity during high caloric intake |
[71] |
I would also encourage the authors to refer to a broader body of work and not restrict some of the sections to their observations/previous publications. For example 5.2. Chronic kidney disease and 5.4. Colorectal cancer
Response: Thank you for this comment. As per your suggestion, we included a broader body of studies in our revised manuscript as follows:
“With regard to the risk of CKD in MHO subjects, prior studies have reported contradictory results [97, 99-101].” (line 303 - line 369-304 in the revised manuscript)
“Obesity is a well-known risk factor for CRC, however, only few studies investi-gated whether obese patients without metabolic abnormalities are at increased risk of CRC. A prospective cohort study in Korea showed that MHO phenotype is a risk factor for CRC in men [106]. However, recently, Cao et al. used data derived from 390,575 adults from the UK Biobank and reported that even in metabolically healthy individu-als, obesity was associated with increased risks of colorectal cancers [105].” (line 341 - line 346 in the revised manuscript)
Table 3. Clinical outcomes of metabolically healthy obesity versus unhealthy obesity.
|
Outcome |
HR (95% CI) for MHO |
HR (95% CI) for MUO |
Reference |
|
Mortality |
1.81 (1.16–2.84) |
2.01 (1.43–2.83) |
[93] |
|
0.86 (0.79–0.93) |
0.96 (0.91–1.01) |
[94] |
|
|
0.98 (0.87–1.10) |
1.24 (1.16–1.32) |
[87] |
|
|
Cardiovascular events |
1.45 (1.20–1.70) |
2.31 (1.99–2.69) |
[95] |
|
|
1.49 (1.45–1.54) |
2.05 (1.94–2.16)1 2.41 (2.25–2.58)2 2.91 (2.68–3.18)3 |
[96] |
|
1.14 (1.05–1.24) |
1.55 (1.47–1.63) |
[94] |
|
|
Chronic kidney disease |
0.83 (0.36–1.70) |
2.80 (1.45–5.35) |
[112] |
|
1.23 (1.12–1.36) |
1.98 (1.85–2.10) |
[83] |
|
|
|
1.17 (0.89–1.53) 2.21 (1.59–3.08) 2.20 (1.55–3.11) |
2.25 (1.91–2.65)4 2.75 (2.32–3.25)5 4.02 (3.40–4.75)6 |
[99]7 |
|
|
1.52 (0.93–2.49) |
2.20 (1.44–3.38) |
[100] |
|
|
0.95 (0.49–1.83) (men) 0.95 (0.74–1.20) (women) |
2.22 (1.44–3.41) (men) 1.23 (1.01–1.54) (women) |
[101] |
|
Alzheimers’ disease |
0.73 (0.54–0.97) |
0.93 (0.70–1.24) |
[97] |
|
0.73 (0.65-0.81) |
0.96 (0.90-1.03) |
[96] |
|
|
Colorectal cancer |
1.14 (1.04–1.26) |
1.21 (1.13–1.29) |
[107] |
|
|
1.10 (0.92–1.33) |
1.29 (1.14–1.47) |
[105] |
|
|
1.21 (1.06–1.39) (men) 1.10 (0.94–1.28) (women) |
1.32 (1.19–1.48) (men) 1.08 (0.95–1.23) (women) |
[106] |
CI, confidence interval; HR, hazard ratio; MHNO, metabolically healthy nonobesity; MHO, metabolically healthy obesity; MUO, metabolically unhealthy obesity.
1Obesity with 1 metabolic risk factor; 2Obesity with 2 metabolic risk factors; 3Obesity with 3 metabolic risk factors; 4class I obesity, body mass index (BMI) 30–34.9 kg/m2; 5class II obesity, BMI 35–39.9 kg/m2; 6class III obesity, BMI ≥40 kg/m2; 7HR for kidney function decline defined as eGFR decline ≥30%;
In general the quality of this review is good but would benefit a lot by the suggestions above.
Response: We sincerely appreciate your insightful comments which substantially improved our manuscript.

Round 2
Reviewer 1 Report
The authors have very well addressed the critical points.